# Maturation of selected human mitochondrial tRNAs requires deadenylation

Sarah F Pearce[1†], Joanna Rorbach[1†], Lindsey Van Haute[1], Aaron R D'Souza[1], Pedro Rebelo-Guiomar[1,2], Christopher A Powell[1], Ian Brierley[3], Andrew E Firth[3], Michal Minczuk[1]*

[1]MRC Mitochondrial Biology Unit, University of Cambridge, Cambridge, United Kingdom; [2]Graduate Program in Areas of Basic and Applied Biology, University of Porto, Porto, Portugal; [3]Department of Pathology, University of Cambridge, Cambridge, United Kingdom

**Abstract** Human mitochondria contain a genome (mtDNA) that encodes essential subunits of the oxidative phosphorylation system. Expression of mtDNA entails multi-step maturation of precursor RNA. In other systems, the RNA life cycle involves surveillance mechanisms, however, the details of RNA quality control have not been extensively characterised in human mitochondria. Using a mitochondrial ribosome profiling and mitochondrial poly(A)-tail RNA sequencing (MPAT-Seq) assay, we identify the poly(A)-specific exoribonuclease PDE12 as a major factor for the quality control of mitochondrial non-coding RNAs. The lack of PDE12 results in a spurious polyadenylation of the 3' ends of the mitochondrial (mt-) rRNA and mt-tRNA. While the aberrant adenylation of 16S mt-rRNA did not affect the integrity of the mitoribosome, spurious poly(A) additions to mt-tRNA led to reduced levels of aminoacylated pool of certain mt-tRNAs and mitoribosome stalling at the corresponding codons. Therefore, our data uncover a new, deadenylation-dependent mtRNA maturation pathway in human mitochondria.

*For correspondence: mam@mrc-mbu.cam.ac.uk

Present address: †Department of Medical Biochemistry and Biophysics, Karolinska Institutet, Solna, Sweden

Competing interests: The authors declare that no competing interests exist.

## Introduction

The human mitochondrial genome (mtDNA) encodes a subset of the structural polypeptides required for oxidative phosphorylation (OxPhos). The mitochondrial (mt-) mRNAs are transcribed and then translated within the mitochondrial matrix by highly specialised machinery. The RNA components of the intra-mitochondrial translation system, two mt-rRNAs and 22 mt-tRNAs, are also expressed from mtDNA. However, all protein components required for mtDNA expression are encoded by the nuclear genome and, upon translation on cytoplasmic ribosomes, are imported into mitochondria. In human mitochondria, mature mt-mRNA species carry a non-templated stretch of poly(A) residues of an average length of 45–55 nucleotides, with the exception of ND6, which does not contain a poly(A) tail. Adenylation of mt-mRNA is performed by mitochondrial poly(A) polymerase (mtPAP, also known as PAPD1) (*Nagaike et al., 2005*; *Tomecki et al., 2004*). Notably, disruption of RNA polyadenylation in human mitochondria, due to mutations in mtPAP, has been linked to a neurodegenerative disease (*Crosby et al., 2010*; *Wilson et al., 2014*).

During eukaryotic gene expression, polyadenylation of mRNAs is required for export of mature mRNAs from the nucleus, transcript stability and the ability of mRNAs to be translated. In contrast, in prokaryotes polyadenylation of RNA species generally serves as a signal for degradation (*Rorbach et al., 2014*). For 7 of the 13 human mt-mRNAs (ND1, ND2, ND3, ND4, CytB, COIII, and ATP6) 3' polyadenylation is essential for completion of stop codons and, therefore, the open-reading

frame. However, the requirement for longer poly(A) tails on most mt-mRNAs in human mitochondria is currently not fully understood.

Previously, ourselves and others have identified phosphodiesterase 12 (PDE12) as a mitochondrial protein of the exonuclease/endonuclease/phosphate (EEP) family (*Poulsen et al., 2011*; *Rorbach et al., 2011*). Recombinant PDE12 exhibited a 3′→5′ exoribonuclease activity on poly(A) homopolymers, and to a lesser extent on poly(U) homopolymers. It also specifically removed poly(A) tails from synthetic ND1 and COII transcripts, stopping 6–8 nt upstream of the polyadenylation site (*Rorbach et al., 2011*). Consistent with the in vitro data, when transgenic PDE12 was overexpressed in cultured cells poly(A) tails on mature mt-mRNAs were shortened. This shortening of poly(A) tails led to changes to steady-state levels of mt-RNAs in a transcript-specific manner, with deadenylation leading to an increase in steady-state levels of some transcripts (e.g. ND1, ND2) and a decrease of others (e.g. COI, COII). This suggested that the status and extent of polyadenylation of mt-mRNAs regulates the steady-state level of some mt-mRNAs. In addition, upon prolonged PDE12 overexpression, and associated loss of mt-mRNA polyadenylation, a general defect in mitochondrial translation was observed. Interestingly, overexpression of mitochondrially-targeted cytosolic deadenylase PARN also caused strong inhibition of intra-mitochondrial protein synthesis (*Wydro et al., 2010*). Although this may indicate that polyadenylation is also required for translation of mt-mRNAs (with the exception of ND6), analysis of mt-mRNAs revealed that loss of polyadenylation was accompanied with a loss of stop codons for transcripts lacking 3′ UTRs, as their completion relies on polyadenylation. This removal of the stop codons complicated the assertion that shortening of poly(A) tails led directly to the observed translation defect. Additionally, ablation of a mitochondrial RNA-binding protein LRPPRC in a mouse model resulted in deadenylation of mt-mRNA with no universal effect on translation, suggesting that translation of some mammalian mitochondrial transcripts can be effective even in the absence of a poly(A) tail (*Ruzzenente et al., 2012*).

In order to further investigate the role of PDE12 and RNA polyadenylation in mitochondrial gene expression, we ablated PDE12 in human cells. Unexpectedly, the absence of the PDE12 poly(A)-specific exoribonuclease did not have any effect on the mitochondrial messenger RNA poly(A) tail length, nor its stability. Instead, we observed that PDE12 was required to remove spurious adenylation of the 3′ end of 16S mt-rRNA and mt-tRNAs. While the aberrant adenylation of 16S did not affect the integrity of the mitochondrial ribosome (mitoribosome), next-generation RNA sequencing-assisted ribosome profiling of PDE12 knock-out cells revealed mitoribosome pausing at specific codons leading to a strong mitochondrial translation defect. Subsequent analysis revealed substantially lower levels of certain aminoacylated mt-tRNAs as a result of spurious adenylation of their 3′ ends. We conclude that a unique deadenylation pathway exists in human mitochondria, which is required for maintaining the proper pool of mt-tRNAs by removing 3′ end poly(A) tails after promiscuous adenylation.

## Results

### PDE12 is required for efficient mitochondrial gene expression

In order to investigate the effects of the absence of PDE12, we produced a gene knockout in HEK293 cells using genome editing by a zinc finger nuclease (ZFN). We generated a heterozygous cell line which contained a single indel at the PDE12 gene locus (PDE12+/−) and a PDE12−/− cell line containing two indels, which produced protein-terminating frameshifts. Western blotting confirmed the absence of PDE12 protein expression (*Figure 1A* and *Figure 1—figure supplement 1A–B*). We observed a reduction in the steady-state levels of marker OxPhos subunits for complexes I, III and IV in PDE12−/− suggesting a combined defect in OxPhos complex synthesis and/or assembly in the absence of PDE12 (*Figure 1A*). Consistent with this, the cellular respiration was impaired in cells lacking PDE12 (*Figure 1B*). We further confirmed the compromised OxPhos performance by culturing cells in media containing galactose as the sole carbon source, forcing them to rely more on mitochondrial ATP production. Under these conditions, the growth rate of PDE12−/− cells was greatly reduced as compared to controls (*Figure 1C*). We also detected a reduction in mitochondrial translation in PDE12−/− as compared to either PDE12+/+ or PDE12+/− (*Figure 1D*). Quantification of a subset of [35S]Met-labeled translation products indicated that longer mitochondrial translation products were affected to a greater degree than the shorter products (*Figure 1E*). This indicated that

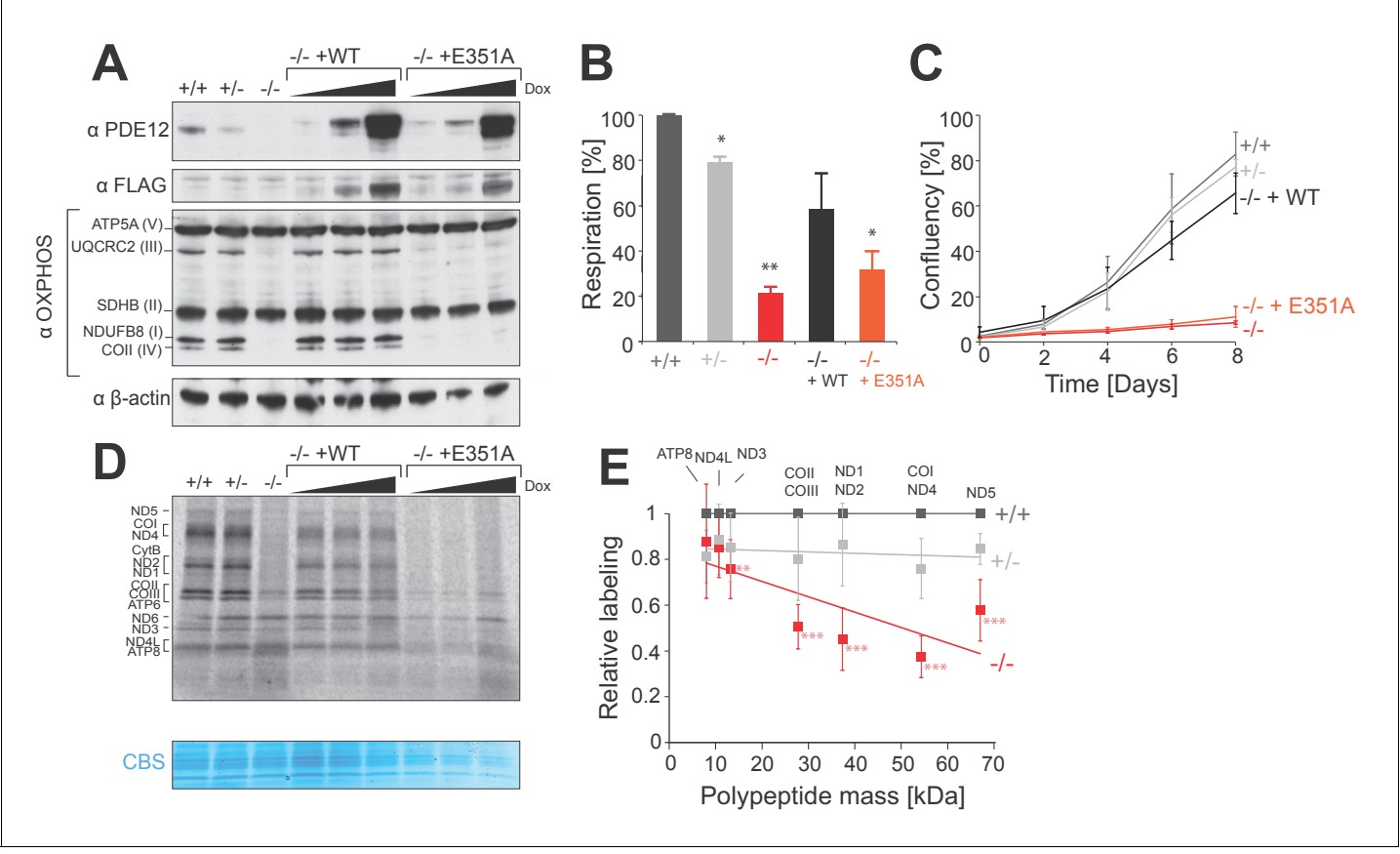

**Figure 1.** PDE12 is essential for efficient mitochondrial translation. (**A**) Western blotting for steady-state levels of OxPhos components in PDE12+/+, PDE12 ± and PDE12−/− cells, and in PDE12−/− expressing either wild-type PDE12 or the E351A catalytic mutant at increasing expression levels. (**B**) Basal oxygen consumption of PDE12+/+ and PDE12−/− cells, and PDE12−/− cells expressing either wild-type PDE12 or E351A. Mean values ± SEM are shown, n = 3. *p<0.05, ***p<0.001 with Student t-test relative to PDE12+/+. (p values: [+/+] vs. [+/−] p=0.0141; [+/+] vs. [−/−] p=0.00134; [+/+] vs. [−/− + WT] p=0.12; [+/+] vs. [−/− + E351A] p=0.0144). (**C**) Growth curve of PDE12+/+, PDE12+/− and PDE12−/− cells, and PDE12−/− cells expressing either PDE12 wild-type or E351A cultured in DMEM containing 0.9 g/L galactose. Representative experiment is shown where each cell line was plated in quadruplicate (mean ± 1 SD). (**D**) Metabolic labelling of mitochondrial translation products with [35S]-methionine in PDE12+/+, PDE12+/− and PDE12−/− cells, and in PDE12−/− cells expressing either wild-type PDE12 or E351A for 24 hr at increasing levels. CBS: Coomassie blue stained gel as loading control. (**E**) Quantification of a subset of mitochondrial translation products in PDE12+/− and PDE12−/− relative to PDE12+/+using ImageQuant software. Relative quantification for each polypeptide was plotted against expected length of polypeptide. For COII/III, ND1/ND2 and COI/ND4 products were quantified together due to proximity of their bands on the gel, and plotted mass is the mean average of both proteins (mean ± SD; n = 6, for PDE12+/− n = 3, **p<0.01, ***p<0.001). (p values: [+/+] vs. [+/−]: ATP8 p=0.148, ND4L p=0.407, ND3 p=0.299, COX2/COX3 p=0.263, ND1/ND2 p=0.397, COX1/ND4 p=0.12124, ND5 p=0.08528; [+/+] vs. [+/+]: ATP8 p=0.328, ND4L p=0.058, ND3 p=0.00857, COX2/COX3 p=9.21×10$^{-5}$, ND1/ND2 p=0.00029, COX1/ND4 p=2.08×10$^{-5}$, ND5 p=0.00016.

The following figure supplement is available for figure 1:

**Figure supplement 1.** Gene targeting of *PDE12* with ZFNs and complementation.

either the initiation of longer mt-mRNAs is specifically affected or that the process of translation elongation is perturbed.

In order to confirm the specificity of phenotypes observed in the PDE12−/− cell line, we generated four further cell lines where *PDE12* was disrupted. Similarly to the initially analysed clonal cell line (*Figure 1*), in these independently constructed PDE12 knock-out cell lines we also observed reductions in the steady-state levels of OxPhos components and in the rate of mitochondrial translation (*Figure 1—figure supplement 1*). Next, we used the Flp-In T-REx system to allow inducible expression of either the wildtype (WT), a catalytically-inactive mutant form of PDE12 (E351A) (*Figure 1A*) or mutants lacking the mitochondrial targeting sequence (Δ16 and Δ23) that are not

imported to mitochondria as confirmed by immunofluorescence analysis (*Figure 1—figure supplement 1E*). As strong and prolonged overexpression of PDE12 had previously been shown to cause a defect in mitochondrial gene expression (*Rorbach et al., 2011*), we employed a range of doxycycline concentrations (0, 0.5, 1.0 ng/ml) in order to titrate levels of PDE12 re-expression. Re-expression of WT PDE12 cDNA, but not the E351A, Δ16 and Δ23 mutants resulted in a rescue of the steady-state levels of OxPhos subunits, growth phenotypes and intra-mitochondrial translation (*Figure 1A–D*, *Figure 1—figure supplement 1F*). This indicated that catalytically active, mitochondrially localised, PDE12 is required for efficient mitochondrial gene expression. As the negative effects on mitochondrial gene expression in PDE12−/− were not accompanied by a decrease in mtDNA copy (*Figure 2—figure supplement 1A*), it was concluded that a defect of either mitochondrial transcription or a post-transcriptional process was occurring in the absence of PDE12.

## *Ablation* of PDE12 does not change mitochondrial messenger RNA steady-state level nor poly(A) tail length

We next sought to determine whether alterations to the mitochondrial transcriptome have led to the mitochondrial translation defect observed in the PDE12−/− cells. As PDE12 had previously been characterised as a poly(A)-specific mitochondrial exoribonuclease and its overexpression led to a decrease of poly(A) tail length and altered steady-state level of mt-mRNAs (*Rorbach et al., 2011*), we set out to examine the effects of absence of PDE12 on mt-mRNAs. Our previously published data showed that under conditions of PDE12 overexpression, poly(A) tails on the ND1 and COII mt-mRNAs were shortened (*Rorbach et al., 2011*). However, in the absence of PDE12, no significant change to poly(A) tail length of either ND1 or COII was observed, when assayed either via mitochondrial poly(A) tail assay (MPAT) or circularisation RT-PCR (cRT-PCR) (*Figure 2A–B*). Furthermore, no change in poly(A) length was observed for all further polyadenylated mt-mRNAs, assessed via northern blotting or MPAT (*Figure 2C* and *Figure 2—figure supplement 1B*). Northern blotting also demonstrated the steady-state levels of all mt-mRNAs were not greatly reduced in PDE12−/− (*Figure 2C* and *Figure 2—figure supplement 1C*). This observation was confirmed by the application of RNAseq to determine differential expression of mt-mRNAs in PDE12−/− (*Figure 2D*). RNA-seq also confirmed that anti-sense transcripts were not accumulating in PDE12−/− (*Figure 2D*). Together these data suggest that neither changes in steady-state level or poly(A) tail length of mt-mRNAs are likely to be the underlying cause of the severe mitochondrial translation defect observed in PDE12−/− cells (*Figure 1D*).

## Translating mitoribosomes stall on specific codons in the absence of the PDE12 deadenylase

We set out to further examine the cause for the mitochondrial translation defect observed in PDE12−/− cells. Firstly, we performed sucrose-gradient fractionation of PDE12+/+ and PDE12−/− total cell lysates to determine if mt-mRNAs were correctly loaded on mitochondrial monosomes. We monitored migration of the 28S small (mt-SSU) and 39S large (mt-LSU) mitochondrial ribosomal subunits by western blotting for the mS18b and bL12 mitochondrial ribosomal proteins (MRPs), respectively (*Figure 3A*). Migration of the mt-SSU and mt-LSU were further confirmed via northern blotting for the 12S and 16S mt-rRNAs (*Figure 3A*). This analysis indicated that in both PDE12+/+ and PDE12−/−, the 55S monosome was formed correctly, migrating in fractions 12–15 of the sucrose gradient (*Figure 3A*). Northern blotting of mt-mRNAs indicated proper loading of ND1, ND2, COI and COII transcripts on the 55S monosome in PDE12−/− (*Figure 3A*), despite severe reduction in their translation (*Figure 1D–E*). Combined MRP western blotting and mt-mRNA and mt-rRNA northern blotting suggested accumulation of mitochondrial transcripts in the monosome fractions in PDE12−/− as compared to PDE12+/+ (*Figure 3A*). Together with the previous observation that translation of longer transcripts was more severely compromised (*Figure 1E*), we next sought to determine whether the observed accumulation of mt-mRNA in the 55S monosome fractions was the result of stalled elongating mitoribosomes. We employed mitochondrial ribosome profiling to determine if elongating mitoribosomes in the PDE12−/− cells were stalling at specific points during translation. This approach detected highly increased incidence of mitochondrial ribosomal protected fragments (mtRPFs) centred on AAA and AAG codons (*Figure 3B*). Similarly to the universal genetic code, in human mitochondria AAA and AAG encode lysine (Lys). As a control for this analysis, we

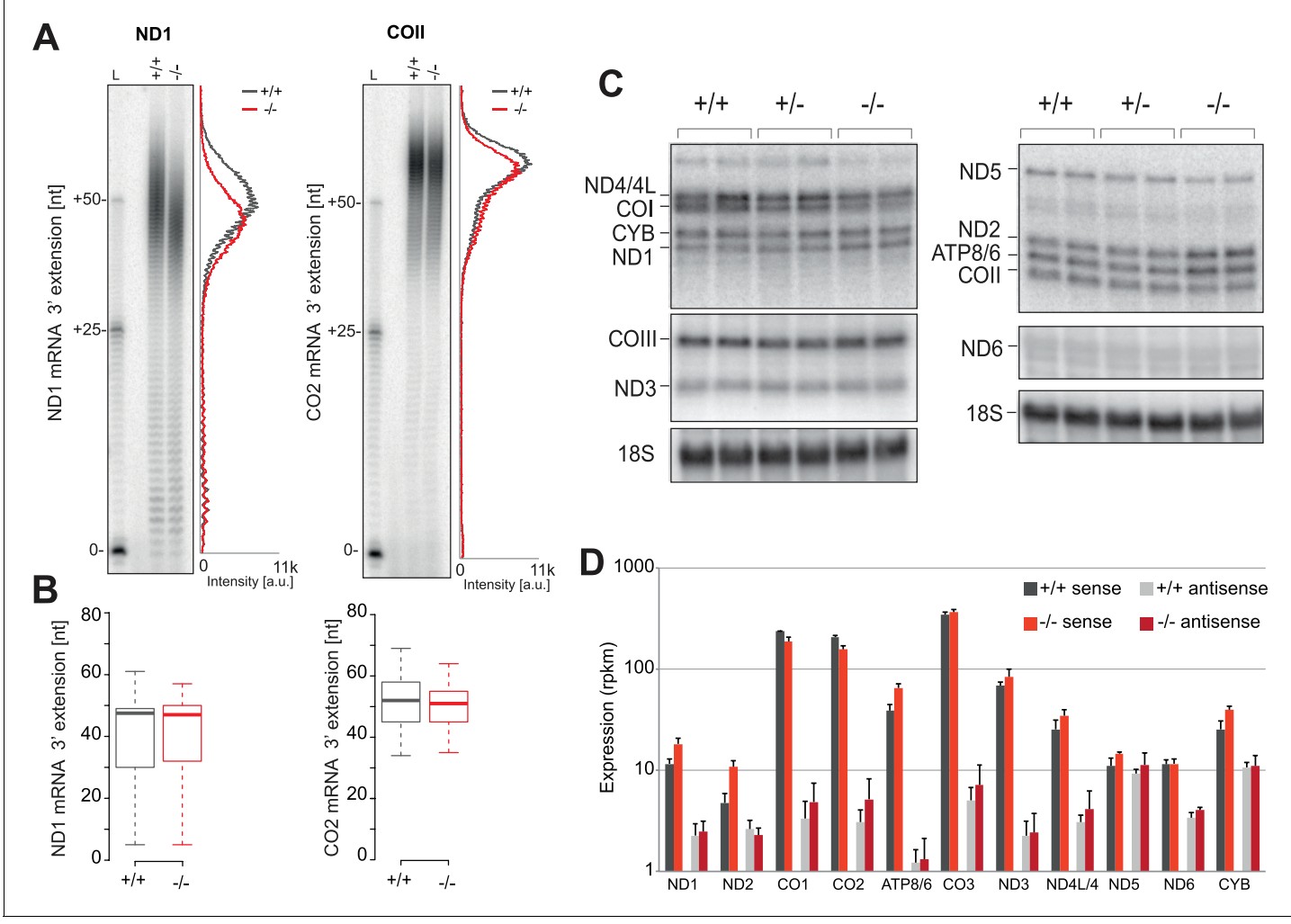

**Figure 2.** Mitochondrial mRNAs are unaffected by absence of PDE12. (**A**) Radioactive MPAT assay for ND1 and COII transcripts for PDE12+/+ and PDE12−/− with densitometric profile of tail length determined via ImageQuant. (**B**) Box plot representing length of 3' extensions to ND1 and COII transcripts for PDE12+/+ and PDE12−/− determined via circularisation RT-PCR. PDE12−/− not significant with Student t-test relative to PDE12+/+. (p values: COX2 p=0.63, ND1 p=0.99). (**C**) Northern blots of mt-mRNA for PDE12+/+, PDE12+/− and PDE12−/−. Nuclear-encoded 18S rRNA was used as a loading control. Quantification provided in *Figure 2—figure supplement 1C*. (**D**) Steady-state levels of mt-mRNAs and mt-mRNA antisense transcripts for PDE12+/+, and PDE12−/− assessed via RNA-seq.

The following figure supplement is available for figure 2:

**Figure supplement 1.** mtDNA copy-number and mt-mRNA are unaffected in PDE12−/−.

performed mitochondrial ribosomal profiling using the ΔFLP 143B cybrid cell line containing mtDNA, which carries a homoplasmic deletion between m.7,846 and m.9,748, resulting in the loss of one mt-tRNA, mt-tRNA[Lys], in addition to a number of mt-mRNA genes (COII, ATP8/6 and COIII) (*Gilkerson et al., 2008*). This analysis revealed a comparable profile for accumulation of mtRPFs centred on AAA and AAG Lys codons as observed for PDE12−/− (*Figure 3—figure supplement 1*). To further corroborate the observation of specific mitoribosome stalling on Lys codons, we analysed the mtRPF profile of the bicistronic ND4L/ND4 mt-mRNA in more detail. Translation of the ND4L/ND4 transcript results in the synthesis of both encoded proteins through separate initiation events. Importantly, the ND4L ORF does not contain any Lys codons, while the ND4 ORF contains 11 Lys codons. While synthesis of the shorter ND4L protein was not significantly altered in PDE12−/− compared to PDE12+/+, translation of the longer ND4 protein was severely reduced in PDE12−/−

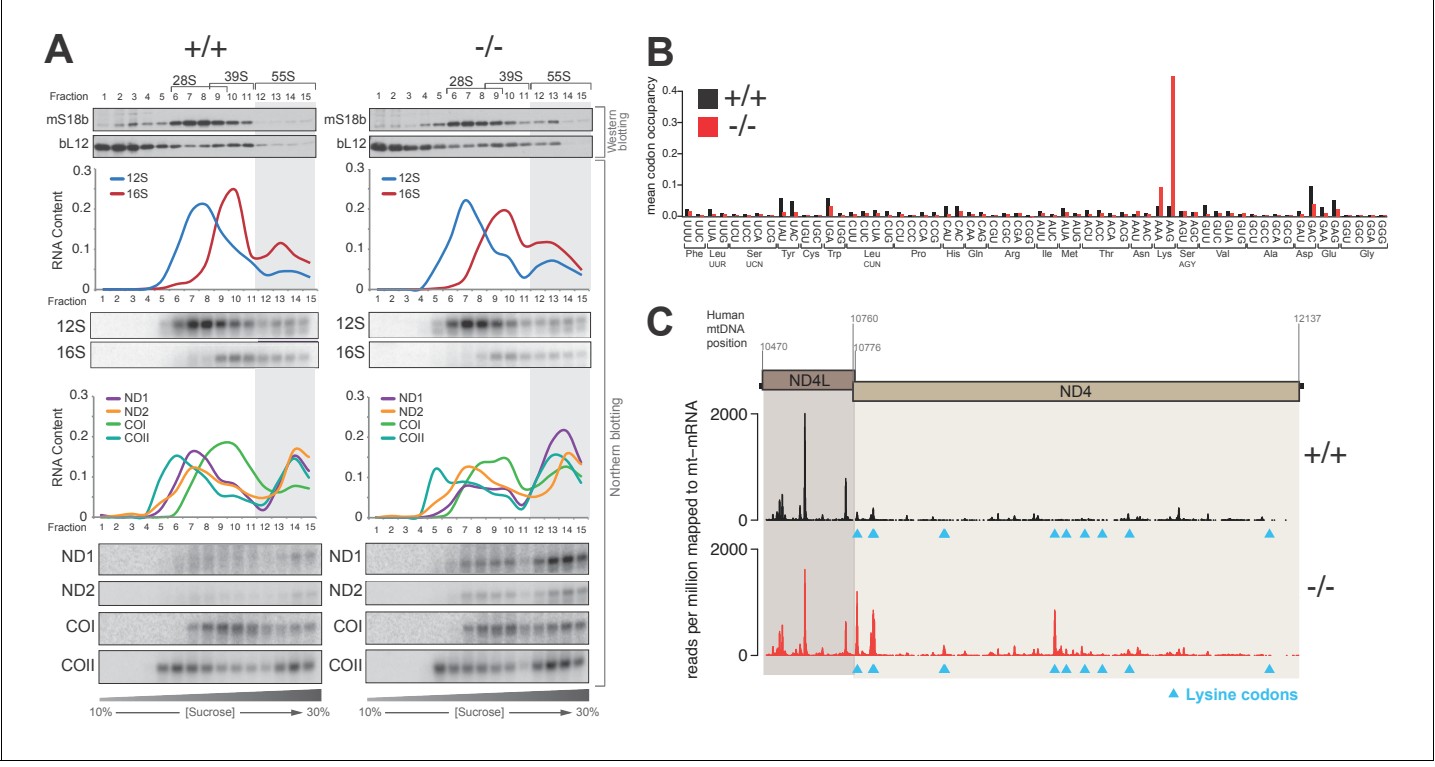

**Figure 3.** Knockout of PDE12 leads to increased stalling events on selected codons on mtDNA-encoded mRNAs. (**A**) Sedimentation of mitochondrial ribosomes on 10–30% isokinetic sucrose gradients for PDE12+/− and PDE12−/− cell lines. Fractions obtained for PDE12+/+, and PDE12−/− were simultaneously analysed by western blotting with antibodies to bL12 (mt-LSU) and mS18b (mt-SSU). Northern blotting was performed using RNA extracted from each fraction with the indicated probes. Quantification of relative RNA content in each fraction was performed using ImageQuant. (**B**) Mean A-site codon occupancy of mitochondrial ribosomal protected fragments (mtRPFs) in the mitochondrial transcriptome of PDE12+/+ and PDE12−/−. See Materials and methods for further explanation. (**C**) mtRPF profile for the ND4L/ND4 transcript in PDE12+/+ and PDE12−/− as an example of increased stalling of mitoribosomes at lysine codons (Note: The ND4L ORF does not contain any lysine codons).

The following figure supplement is available for figure 3:

**Figure supplement 1.** Increased stalling events on lysine codons in ΔFLP cell line.

(**Figure 1E**). The mtRPF profile across the ND4L ORF was highly similar between PDE12+/+ and PDE12−/−, consistent with the lack of any Lys codons and no effect on intra-mitochondrial translation rate for ND4L (**Figure 3C**). In contrast, distinct mtRPF peaks were present along the ND4 ORF in PDE12−/−, each corresponding to the location of a Lys codon (**Figure 3C**). Taken together, we concluded that in the absence of PDE12 the mitoribosome is assembled correctly, with mitochondrial messengers being loaded on the 55S monosome, however, specific mitoribosomal stalling occurs, predominantly on Lys codons.

## PDE12 is required for removal of spurious 3' adenylation from mt-tRNAs

Given that we previously characterised PDE12 as a poly(A)-specific exoribonuclease with 3'→5' directionality and the observation of specific mitoribosome stalling mainly at Lys codons, we next decided to examine the 3' ends of mt-tRNA[Lys] in PDE12−/− (**Figure 4A**). Radioactive MPAT revealed that in PDE12−/− mt-tRNA[Lys] harboured 6–9 nt 3' extensions (**Figure 4B** and **Figure 4— figure supplement 1**). Expression of catalytically active PDE12, but not the E351A mutant, restored normal 3' ends of mt-tRNAs (**Figure 4—figure supplement 1**). In order to determine the primary RNA sequence of the 3' additions to mt-tRNAs in PDE12−/−, as well as to determine the transcriptome-wide effect of PDE12 inactivation on all mt-tRNAs, we employed MPAT combined with deep

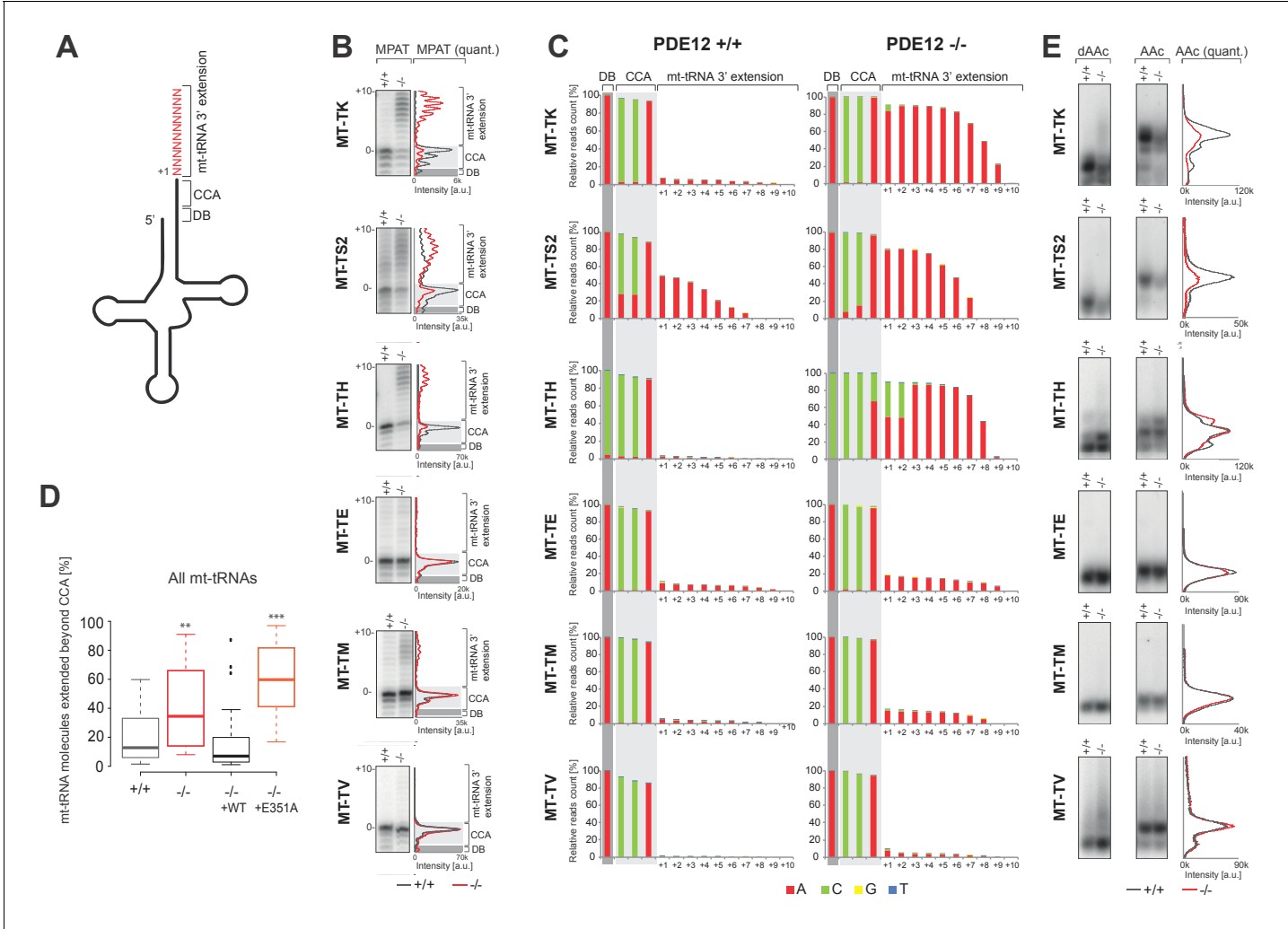

**Figure 4.** A subset of mt-tRNAs are affected by spurious 3' adenylation in the absence of PDE12, leading to reduced aminoacylation of specific mt-tRNAs. (**A**) Schematic of mt-tRNA structure depicting 3' extension determined via the radioactive MPAT (**b**) and MPAT-Seq (**c**). DB, discriminator base (**B**) Radioactive MPAT assay for a subset of mt-tRNAs extracted from PDE12+/+ and PDE12−/−. MPAT assay gel profiles were determined using ImageQuant. (**C**) Representation of 3' ends of a subset of mt-tRNAs from PDE12+/+ and PDE12−/−, ascertained by MPAT-Seq. Read count shown for each position is relative to the read count for the corresponding discriminator base (DB) for each mt-tRNA. (**D**) Box-plot representation of percentage of mt-tRNAs molecules extended beyond the 3' CCA addition for all 22 mt-tRNAs, ascertained by MPAT-Seq, for PDE12+/+ and PDE12−/− cells, and in PDE12−/− cells expressing either wild-type PDE12 or E351A for 24 hr. **p<0.01, ***p<0.001 with Student t-test relative to PDE12+/+. (p values: [+/+] vs. [−/−] p=0.0089, [+/+] vs. [−/− + WT] p=0.83, [+/+] vs [−/− + E351A] p=2.14×10⁻⁷). (**E**) Aminoacylation assay for subset of mt-tRNAs for PDE12+/+ and PDE12−/−. Profiles of aminoacylated mt-tRNAs were determined using ImageQuant. dAAc indicates intentionally deacylated samples.

The following source data and figure supplements are available for figure 4:

**Source data 1.** Read count for mitochondrial poly(A) (MPAT) next generation RNA sequencing (MPAT-Seq).
**Figure supplement 1.** Radioactive MPAT assay for mt-tRNA^Lys.
**Figure supplement 2.** Northern blotting for mt-tRNAs.
**Figure supplement 3.** Aminoacylation analysis by sodium periodate oxidation.

RNA sequencing (MPAT-Seq). This analysis revealed that the extended 3' ends of mt-tRNA$^{Lys}$ (MT-TK) were the result of spurious adenylation on the 3' ends of mature mt-tRNA molecules following CCA addition (**Figure 4C**). In addition, MPAT-Seq also revealed other mt-tRNAs were affected by spurious 3' adenylation in PDE12−/−, notable examples being mt-tRNA$^{Ser(AGY)}$ (MT-TS2) and mt-tRNA$^{His}$ (MT-TH), while others remained largely unaffected (**Figure 4C**, **Figure 4—source data 1**). We corroborated the results of MPAT-Seq with conventional radioactive MPAT analysis for selected mt-tRNAs (mt-tRNA $^{Ser(AGY),\ His,\ -Glu,\ -Met,\ -Val}$), revealing a good agreement between the two approaches (**Figure 4B**). Global analysis of the 3' end extended proportion of molecules for each mt-tRNA revealed a general increase in the proportion of affected molecules in PDE12−/− (**Figure 4D**). The proportion of extended mt-tRNA molecules was reduced upon re-expression of catalytically active PDE12, but not the E351A mutant (**Figure 4D**). High-resolution northern blotting showed that oligoadenylation of mt-tRNA in PDE12−/− cells leads to a significant reduction of mature mt-tRNA$^{Lys}$ available for aminoacylation, presumably due to the large proportion of molecules extended on the 3' end (**Figure 4B–C**). The total steady-state levels of mature and oligoadenylated mt-tRNA$^{Lys}$ remained unchanged (**Figure 4—figure supplement 2**). As CCA addition is required for aminoacylation of mt-tRNAs, it was expected that 3' extensions to the CCA would result in a reduction in the proportion of mt-tRNAs available for aminoacylation. Indeed, an aminoacylation assay based on the usage of low pH to distinguish between the aminoacyl-tRNA and the uncharged tRNA revealed severe reduction in the overall amount of charged mt-tRNA$^{Lys}$ (**Figure 4E**). An alternative method for the measurement of tRNA charging based on periodate oxidation confirmed that only the correctly matured mt-tRNA$^{Lys}$ could be aminoacylated (**Figure 4—figure supplement 3**). We also detected a reduction in the steady-state levels of mature mt-tRNA$^{Ser(AGY)}$ in PDE12−/− (**Figure 4E**, **Figure 4—figure supplement 2**). However, despite the presence of spurious 3' adenylation of mt-tRNA$^{His}$ in PDE12−/−, we did not detect any substantial reduction of the overall level of aminoacylation of this mt-tRNA (**Figure 4E**). The amount of charged molecules for other mt-tRNAs remained unaffected (**Figure 4E**). Taken together, we concluded that PDE12 is necessary for maintaining the integrity of the 3' ends of mt-tRNAs, protecting them from spurious polyadenylation, with the lack of PDE12 leading to accumulation of dysfunctional mt-tRNAs and as a consequence reduced levels of mature mt-tRNAs available for aminoacylation and mitochondrial translation.

## Spurious poly(A) extensions to 16S mt-rRNA do not affect mitochondrial monosome assembly

To determine whether the lack of PDE12 led to spurious extensions on other non-coding mtRNAs, we performed radioactive MPAT (**Figure 5A–B**) as well as cRT-PCR (**Figure 5C–D**) for both 12S and 16S mt-rRNAs. Although the 3' end extensions on 12S were generally heterogenous, they were similar in PDE12+/+ and PDE12−/− (**Figure 5A,C** and **Supplementary file 1**). On the other hand the extensions to the 3' end of 16S, while being usually of 4 or 5 nt in length in control PDE12+/+cells, were significantly longer in PDE12−/− (**Figure 5B,D** and **Supplementary file 1**). The cRT-PCR analysis followed by sequencing also confirmed that the 3' extensions of 16S mt-rRNA were oligoadenylate in nature, similar to those on mt-tRNAs in PDE12−/− (**Supplementary file 1**). Of note, the analysis of 16S 3' ends revealed that they frequently (approximately 50%) contained 4–5 nucleotides encoded by the 5' terminus of the immediately distal mt-tRNA$^{Leu(UUR)}$ (**Supplementary file 1**), suggestive of alternative endonucleolytic processing or alternative transcription termination within the mt-tRNA$^{Leu(UUR)}$ gene. Earlier sucrose-gradient fractionation indicated no assembly defect of either subunit of the mitoribosome (**Figure 3A**), nonetheless, we sought to determine in more detail whether spurious 3' adenylation to 16S mt-rRNA had an effect in the assembly of mt-LSU or 55S monosome. Steady-state levels of both 16S and 12S mt-rRNAs are unaffected in PDE12−/− (**Figure 5E**). In addition, steady-state levels of MRPs were not altered in PDE12−/−, as determined by western blotting (**Figure 5—figure supplement 1**). Of note, mt-LSU components bL17 and bL19 that are adjacent to the 16S 3' end were unaffected. Therefore, the spurious extensions observed on 16S in PDE12−/− do not appear to affect assembly and/or stability of the mt-LSU.

To investigate whether longer 3' extensions on 16S impede the formation of the monosome, we performed 16S MPAT assay for sucrose-gradient fractions known to contain either the free mt-LSU (fractions 9 and 10) or 55S monosome (fractions 12–15). As the extensions of 16S mt-rRNA in the mt-LSU of control PDE12+/+cells are almost exclusively either four or five nt (**Figure 5B–D**), it was possible that only this specific length is compatible with correct monosome assembly. Therefore, in

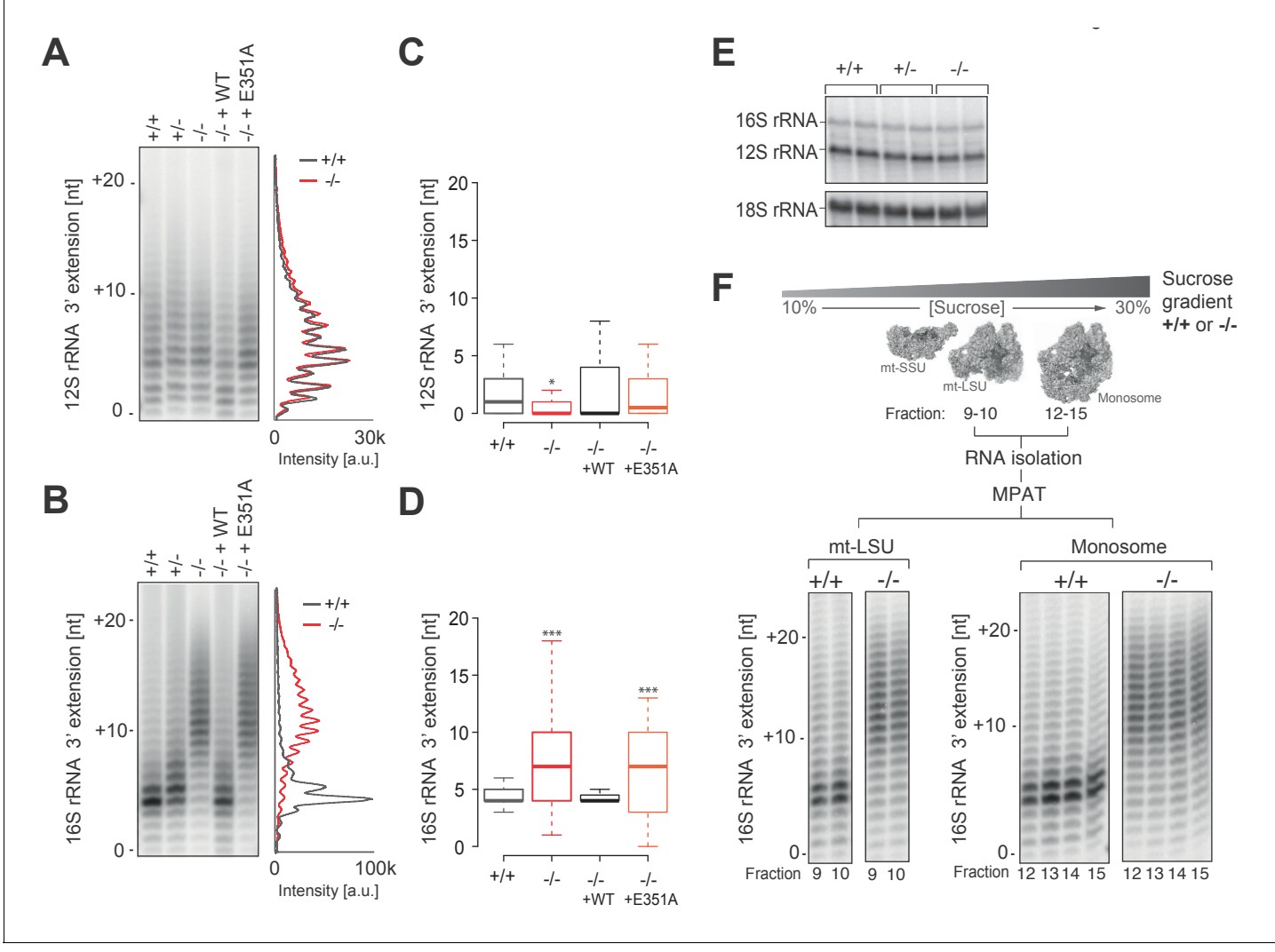

**Figure 5.** Spurious adenylation to the 3' terminus of 16S mt-rRNA in the absence of PDE12 does not affect mitochondrial monosome formation. (**A and B**) Radioactive MPAT assay for 12S (**a**) and 16S (**b**) mt-rRNAs extracted from PDE12+/+, PDE12+/− and PDE12−/− cells, and for PDE12−/− cells expressing either wild-type PDE12 or E351A for 24 hr. (**C and D**) Box-plot representing 3' extensions to 12S (**C**) and 16S (**D**) mt-rRNAs via circularisation RT-PCR for PDE12+/+ and PDE12−/− cells, and from PDE12−/− cells expressing either wild-type PDE12 or E351A. *p<0.05, ***p<0.001 with Student t-test relative to PDE12+/+. (p values: 12S: [+/+] vs. [−/−] p=0.013, [+/+] vs. [−/− + WT] p=0.32, [+/+] vs. [−/− + E351A] p=0.84; 16S: [+/+] vs. [−/−] p=4.6×10⁻⁷, [+/+] vs. [−/− + WT] p=0.53, [+/+] vs. [−/− + E351A] p=5.3×10⁻⁴) (**E**) Steady-state levels of 12S and 16S mt-rRNAs for PDE12+/+, PDE12 +/− and PDE12−/− determined via northern blotting. (**F**) Radioactive MPAT assay for 16S mt-rRNA extracted from sucrose-gradient fractions for free mt-LSU (fractions 9 and 10) and monosome (12-15) for PDE12+/+ and PDE12−/− cells.

The following figure supplement is available for figure 5:

**Figure supplement 1.** Structural and western blot analysis of mtLSU.

PDE12−/− there could be selection pressure against mt-LSU which contained longer 3' extensions from forming 55S particles. However, 3' extensions profiles of 16S mt-rRNA are identical for the free mt-LSU and monosome fractions, indicating that longer 16S 3' extensions do not impede monosome formation (*Figure 5F*). In conclusion, oligoadenylate extensions to 16S mt-rRNA in PDE12−/− do not seem to negatively impact mitoribosome formation and the stalling of mitoribosome in the absence of PDE12 is likely to be predominantly caused by mt-tRNAs harbouring poly(A) extensions.

## Increased adenylation activity is detrimental for mitochondrial gene expression

Mitochondrial poly(A) polymerase (mtPAP, PAPD1) is, thus far, the only identified human mitochondrial enzyme with a poly(A) polymerase activity. Therefore, we hypothesised that mtPAP contributes to the spurious adenylation observed on mt-tRNAs and 16S mt-rRNA (*Figure 4* and *Figure 5*). To determine if an increased expression of mtPAP exaggerates the effect of aberrant 3' adenylation of non-coding mtRNAs, we generated cell lines inducibly expressing untagged human mtPAP protein in either the PDE12+/+ or PDE12−/− backgrounds (*Figure 6A*). Lengthening of poly(A) tails on ND1 and COII was observed in both PDE12+/+ and PDE12−/− upon mtPAP induction (*Figure 6B*). The extent of this increased polyadenylation was similar in PDE12+/+ and PDE12−/−, further supporting the evidence that PDE12 is not required for controlling the length of poly(A) tails of mt-mRNAs. Further MPAT-Seq analysis revealed that overexpression of mtPAP in control PDE12+/+ cells led to an increase in the proportion of mt-tRNAs which carry 3' extensions, with this increase being comparable to the levels observed in PDE12−/− (*Figure 6C* and *Figure 4—source data 1*). However, induction of mtPAP expression in PDE12−/− led to a greatly increased general proportion of extended mt-tRNAs (*Figure 6C* and *Figure 4—source data 1*), suggesting that PDE12 is required to deadenylate the mt-tRNAs erroneously modified by mtPAP. Overexpression of mtPAP also led to further lengthening of 3' extensions of 16S mt-rRNA in PDE12−/−, but not in PDE12+/+, suggesting that 16S mt-rRNA are protected from aberrant adenylation when PDE12 is present at endogenous level (*Figure 6D*). On the other hand, 3' adenylation of 12S mt-rRNA was affected to a similar extent by mtPAP overexpression in either PDE12+/+ or PDE12−/− (*Figure 6E*), supporting the previous notion of PDE12 not being involved in the maturation of 12S mt-rRNA (*Figure 4*). Overexpression of mtPAP in PDE12+/+ led to mild reduction in mitochondrial translation and the steady-state levels of OxPhos subunits (*Figure 6A*), However, it is indistinguishable whether this is a result of increased mt-mRNA poly(A) tail length, 3' extension to 12S mt-rRNA, increased global 3' adenylation to mt-tRNAs, or a combination of these effects (*Figure 6D–E*). Nonetheless, upon mtPAP overexpression in PDE12−/− cells the defect in mitochondrial translation and the steady-state levels of the OxPhos components were exaggerated (*Figure 6A*). This is consistent with further impairment of mt-tRNAs and 16S mt-tRNA 3' ends by spurious polyadenylation when PDE12 is absent (*Figure 6C–D*), providing an additional support for the essential role of PDE12 in the deadenylation of mitochondrial non-coding RNAs upon incorrect modification of their 3' ends.

## Discussion

In mammalian mitochondria, all RNA species are derived from long, polycistronic precursor RNAs following transcription of mtDNA. This implies that the key regulatory mechanisms governing their function must operate post-transcriptionally, yet our knowledge of these mechanisms is far from complete. Here, we discovered a novel role for deadenylation in human mitochondria, which is to remove promiscuous polyadenylaton from mitochondrial non-coding RNAs. Importantly, we show that this process is of significance for the maintenance of a proper pool of aminoacylated mt-tRNA, and in particular mt-tRNA$^{Lys}$. One of the major consequences of the lack of deadenylase activity is the frequent stalling of mitoribosomes, mainly at lysine codons, as revealed by the mitochondrial ribosome profiling approach, with defective intra-mitochondrial translation and OXPHOS occurring as a result (*Figure 6F*).

### Regulation of poly(A) tails of mitochondrial messenger RNA

Poly(A) tails play a major regulatory role in determining the stability and translational efficiency of eukaryotic nuclear-encoded messenger RNAs. Exonucleolytic removal of poly(A) tails (deadenylation) is the rate-limiting step in mRNA decay in the cytosol, with the majority of deadenylation activity contributed by the Ccr4-Not and Pan2-Pan3 complexes (*Passmore and Taatjes, 2016*). However, herein our comprehensive characterisation of mitochondrial messenger RNAs (*Figure 2* and *Figure 2—figure supplement 1*) did not reveal any appreciable changes in poly(A) tail or stability of mt-mRNAs in the absence of the PDE12 deadenylase, previously shown to localise to human mitochondria (*Poulsen et al., 2011*; *Rorbach et al., 2011*). This strongly suggests that PDE12 is not involved in the maintenance and function of mt-mRNA in normal physiological conditions. It is

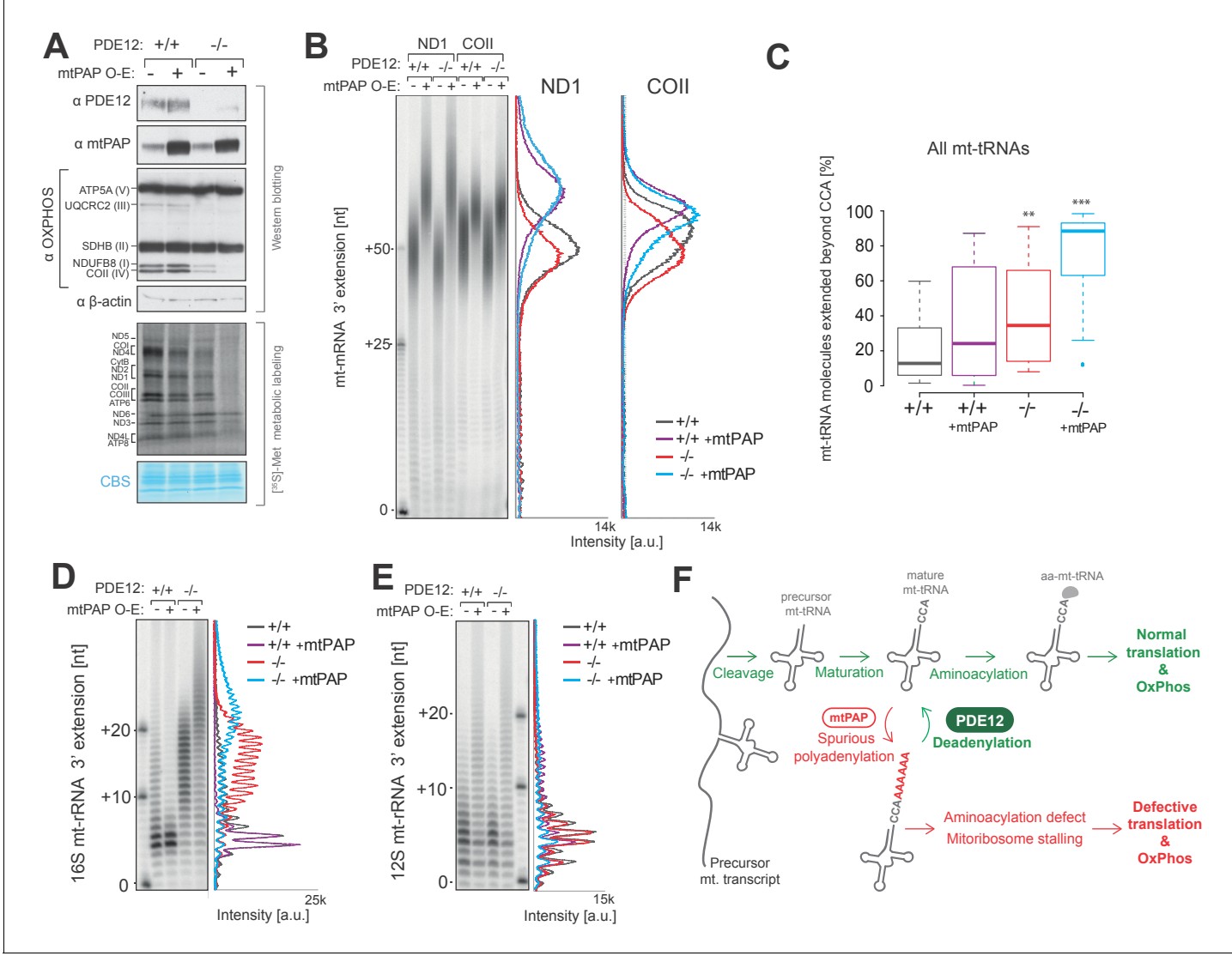

**Figure 6.** Increased adenylation activity is detrimental to mitochondrial gene expression. (A) Western blotting for steady-state levels of OxPhos components and mtPAP (top). Mitochondrial translation products were labelled with [35S]-Met in PDE12+/+ and PDE12−/− cells with either endogenous or overexpressed levels of mtPAP (bottom). (B) Radioactive MPAT assay for ND1 and COII extracted from PDE12+/+ and PDE12−/− cells with either endogenous or overexpressed levels of mtPAP. MPAT assay gel profiles were determined using ImageQuant. (C) Box-plot representation of percentage of mt-tRNAs molecules extended beyond the 3' CCA addition for all 22 mt-tRNAs, ascertained by MPAT-Seq, for PDE12+/+ and PDE12−/− cells with either endogenous or overexpressed levels of mtPAP. **p<0.01, ***p<0.001 with Student t-test relative to PDE12+/+. (p values: [+/+] vs. [+/+ + mtPAP] p=0.053, [+/+] vs [−/−] p=0.0089, [+/+] vs [−/− + mtPAP] p=5.52×10$^{-11}$). (D and E) Radioactive MPAT assay for 16S (D) and 12S (E) mt-rRNAs extracted from PDE12+/+ and PDE12−/− cells with either endogenous or overexpressed levels of mtPAP. MPAT assay gel profiles were determined using ImageQuant. (F) Model of PDE12 function. Polycistronic precursor RNA is endonucleolytically processed, liberating the individual mt-tRNA transcripts, which undergo post-transcriptional maturation (nucleotide modifications, CCA addition) before being aminoacylated with a cognate amino acid for use in mitochondrial translation (green). In the case of some mt-tRNAs a spurious poly(A) tail is added by mtPAP, preventing aminoacylation, leading to mitoribosome stalling during mitochondrial translation and to OxPhos defect as a consequence (red). Deadenylation by PDE12 removes spurious polyadenylation on mt-tRNAs restoring the properly matured pool of mt-tRNAs available for aminoacylation (green). Note: The role of PDE12 in the processing of mt-rRNA not included.

The following figure supplement is available for figure 6:

**Figure supplement 1.** The role of PDE12 in maturation of mt-tRNA$^{Tyr}$.

possible that there are alternative deadenylases regulating the length of poly(A) tails on human mitochondrial protein-coding transcripts. Further studies will be required to elucidate how the length of poly(A) tails is controlled in human mitochondria.

## The role of deadenylation in the biogenesis of mitochondrial tRNA

During tRNA biogenesis, the nucleolytic removal of the 5′ leader and the 3′ trailer does not usually produce the complete tRNA and additional maturation steps of their ends are required. Of the most widespread is the post-transcriptional template-independent addition of CCA at the tRNA 3′ end. Furthermore, all tRNAs are subjected to extensive post-transcriptional nucleotide modifications, with mammalian mt-tRNAs harbouring approximately 7% of mt-tRNA chemically modified nucleotides by dedicated, nDNA-encoded enzymes (*Powell et al., 2015b*; *Suzuki and Suzuki, 2014*). The energetic expenditure by the cell to achieve the full maturation of tRNAs, is presumably one of the reasons why their half-life is generally longer as compared to messenger RNA (*Deutscher, 2006*). This might also justify the existence of tRNA repair mechanism described for other systems; it is advantageous for the cells to repair a partially damaged mt-tRNA rather than re-synthetize and subject it to multi-step maturation. Enzymes responsible for the basic tRNA maturation processes have also been reported to be involved in tRNA repair after damage (*Phizicky, 2008*). For example, it has been suggested that the 3′−5′ reverse RNA polymerase, Thg1, normally involved in the synthesis of $G_{-1}$ in tRNA$^{His}$, has a role in repair of truncated RNA 5′ ends (*Jackman and Phizicky, 2006*). Also, dedicated enzymes such as RNA ligases (*Nandakumar et al., 2008*) or demethylating enzymes (*Ougland et al., 2004*) can directly neutralise the damage to tRNA. Our results herein indicate that mechanisms of maintaining the integrity of tRNA via the deadenylation of matured mt-tRNAs exist in mammalian mitochondria and involve the exoribonuclease PDE12. In this context, the role of PDE12 can be considered not only as a part of the mt-tRNA maturation pathway, but also as a component of mt-tRNA quality control, with a role in the 3′ end repair upon aberrant polyadenylation. It remains to be established whether PDE12-assisted deadenylation of aberrantly polyadenylated mt-tRNAs also plays a regulatory role for mt-tRNA pools. It has been shown that polyadenylation helps to regulate levels of functional tRNAs in *E. coli* (*Mohanty and Kushner, 2013*). Increased expression of the bacterial poly(A) polymerase, PAP I, leads to the rapid polyadenylation of mature tRNAs, reducing the fraction of aminoacylated tRNAs and, therefore, limiting levels of functional tRNAs for protein synthesis. In light of our results, it is possible that mitochondrial polyadenylation by mtPAP may control the levels of functional mt-tRNA by acting on their 3′ ends and that PDE12 might be an important element of such regulatory pathway.

In the eukaryotic nucleus, the TRAMP complex (Trf4/Air2/Mtr4), has been shown to function in marking nuclear non-coding RNAs with oligoadenylation which acts as a docking site for nuclear exosome degradation (*Anderson and Wang, 2009*). This prokaryote-like approach functions to destabilise unrequired nuclear RNAs, including hypomodified initiator tRNA$^{Met}$ (*Wang et al., 2008*), in addition to other non-coding RNAs such as abnormally processed ribosomal RNAs. Hypothetically, PDE12 could function in a similar manner in the complete turnover of aberrantly matured molecules from the mt-tRNA pools in human mitochondria, with the oligoadenylation observed herein functioning as a signal for degradation. However, following PDE12 knockout, we observed no changes in overall steady-state level of total mt-tRNA pools, despite a proportion of some mt-tRNA species carrying oligoadenylation (*Figure 4—figure supplement 2*). This would suggest that the purpose of the oligoadenylation is unlikely to be a signal for complete turnover of the molecule in human mitochondria.

## The promiscuous polyadenylation activity in human mitochondria

Our results suggest that mitochondrial poly(A) polymerase, mtPAP, despite being critical for mitochondrial gene expression through maturation of mt-mRNAs (*Bratic et al., 2016*; *Crosby et al., 2010*; *Wilson et al., 2014*), is a promiscuous enzyme. Although we cannot exclude any other, as yet unidentified, mitochondrial poly(A) polymerase being involved in spurious mtRNA adenylation, here we show that overexpression of mtPAP causes increased levels of aberrantly polyadenylated mtRNA, with the effect being more pronounced in the absence of PDE12 (*Figure 6*). Interestingly, the levels of spurious polyadenylation and its functional effects vary between the different mt-rRNA and mt-tRNA species. While 12S mt-rRNA was not found to be susceptible to promiscuous mtPAP activity in

the absence of PDE12, 16S mt-rRNA contained abnormal 3' end extensions. Nonetheless, the extended 16S mt-rRNA remained functional (**Figure 5**). Similarly, selectivity of spurious adenylation towards different mt-tRNAs was observed, with mt-tRNA$^{Lys}$, $^{-Ser(AGY)}$ and $^{-His}$ being the most affected species. Despite comparable levels of aberrant 3' end polyadenylation of these three mt-tRNAs detected by the radioactive MPAT and MPAT-Seq (**Figure 4B–C**), high-resolution northern blotting (**Figure 4—figure supplement 2**) revealed that spurious adenylation actually affects mt-tRNA$^{Lys}$ to a greater extent. This discrepancy is likely to stem from the intricacies of the MPAT assays (**Figure 4B– D**). As the MPAT assays require reverse transcriptase to read through post-transcriptionally modified bases, the measurements of mt-tRNA adenylation might not be accurate, specifically by over-representing the extent of 3' modification of unmodified molecules relative to that of mature mt-tRNAs. Therefore, the result of the MPAT assays may not exactly reflect the proportion of adenylated molecules in the total pool of each mt-tRNA species.

The steady-state levels of mature mt-tRNA$^{Lys}$ and mt-tRNA$^{Ser(AGY)}$ were substantially decreased, with the levels of mt-tRNA$^{His}$ available for aminoacylation remaining comparable to the control (**Figure 4E**). Notwithstanding with equally diminished pools of aa-mt-tRNA$^{Lys}$ and aa-mt-tRNA$^{Ser(AGY)}$, mitoribosome stalling was predominately detected on Lys codons. The difference in mitoribosomal stalling on Lys and Ser$^{AGY}$ codons might be explained by the relative percentages of their occurrence in mtDNA-encoded ORFs. The Ser$^{AGY}$ codon is the second least frequently occurring coding triplet and accounts for 1.4% of all mtDNA codons, whereas Lys occurs with a frequency of 2.5%.

## Other cellular roles of PDE12

Recently, PDE12 has been proposed to act in the maturation of mt-tRNA$^{Tyr}$. In human mtDNA, the mt-tRNA$^{Tyr}$ and mt-tRNA$^{Cys}$ genes overlap by one nucleotide at their 3' and 5' termini, respectively (**Fiedler et al., 2015**). During endonucleolytic processing, therefore, an 'A' residue, which is shared between the two mt-tRNAs, is missing from the mature form of one of the mt-tRNA species (**Figure 6—figure supplement 1**). There have been conflicting reports, based on in vitro study, regarding the exact nature of the endonucleolytic processing of the mt-tRNA$^{Tyr}$-mt-tRNA$^{Cys}$ junction in human mitochondria, where cleavage results either in the loss of the mt-tRNA$^{Tyr}$ discriminator base (DB), producing a complete mt-tRNA$^{Cys}$ molecule (**Reichert et al., 1998**), or in the retention of the DB by mt-tRNA$^{Tyr}$ (**Rossmanith et al., 1995**). It has also been recently proposed that alternative processing at this site may occur, resulting in pools of complete and incomplete species of each mt-tRNA (**Fiedler et al., 2015**). According to this model, the 3' end of a pool of mt-tRNA$^{Tyr}$ would need to be repaired via adenylation to provide the DB, with mtPAP performing the 3' adenylation of mt-tRNA$^{Tyr}$ that need subsequent trimming. Based on in vitro reconstitution experiments, PDE12 and/ or the mitochondrial RNase Z (ELAC2) have been proposed to trim the polyadenylated mt-tRNA$^{Tyr}$ precursor leaving the 'A' at the DB position for the subsequent CCA addition (**Figure 6—figure supplement 1**). However, our radioactive MPAT and MPAT-Seq assay in control and PDE12−/− cells revealed no changes in the mt-tRNA$^{Tyr}$ 3' end maturation, suggesting that either PDE12 is not involved in this process in living cells or there is substantial redundancy in the mechanisms of mt-tRNA$^{Tyr}$ biogenesis (**Figure 6—figure supplement 1**).

PDE12 has also been proposed to play a role in the 2-5A system functioning in cellular anti-viral defence. Upon viral infection, the production of interferons (IFNs) is stimulated which act to induce expression of 2'−5'-oligoadenylate synthetases (2'−5'-OASs). 2'−5'-OASs catalyse the synthesis of 5' triphosphorylated 2'−5'-linked oligoadenylate polyribonucleotides (2-5As) from ATP. Accumulation of these 2-5As leads to the activation of latent RNase L which, when activated, degrades viral RNA (**Wreschner et al., 1981**). PDE12 has been suggested to control the cellular levels of 2-5As by catalysing their degradation (**Johnston and Hearl, 1987**; **Kubota et al., 2004**; **Wood et al., 2015**). However, as yet, the role of the 2-5A system has not been characterised in human mitochondria, being predominantly based in the cytosol and nucleus, implying that a putative extra-mitochondrial PDE12 pool would have to exist. However, our previous study indicated exclusive mitochondrial localisation of this protein (**Rorbach et al., 2011**), which was confirmed in this work (**Figure 1—figure supplement 1**). Also, the PDE12 variants without the MTS, which localise in the cytosol, did not rescue the depletion in OxPhos subunits in the PDE12−/− cells (**Figure 1—figure supplement 1**), supporting the notion that mitochondrially localised PDE12 is required for its function in mtDNA

gene expression. The prior assignment of PDE12 as having a role in the 2-5A system, therefore, would require further research.

## Concluding remarks

In summary, we have assigned a novel molecular function for PDE12 in the deadenylation-dependent maturation of mitochondrial non-coding RNAs. We have also uncovered several other features of post-transcriptional regulation of human mitochondrial gene expression, including, for example, the promiscuity of mitochondrial poly(A) polymerase (mtPAP) and the tolerance of several non-coding mtRNAs for spurious, non-templated poly(A) extensions (e.g. 16S mt-rRNA or mt-tRNA$^{Ser(AGY)}$). There is recent increasing evidence that perturbation of mtDNA gene expression is often involved to human respiratory chain disorders (*Nicholls et al., 2013*; *Powell et al., 2015a*; *Van Haute et al., 2016*, *2015*), neurodegenerative disorders, such as amyotrophic lateral sclerosis (ALS) (*Wang et al., 2016*) or viral infection (*Karniely et al., 2016*). In this context, our research provides important new knowledge and molecular tools (e.g. mitoribosome profiling and MPAT-Seq) to study these processes, which will help to understand the pathophysiology of these diseases and will be instrumental in designing future mechanism-based treatments.

# Materials and methods

## Materials

For a comprehensive list of antibodies, vectors and oligonuclotides used in this study please refer to accompanying *Supplementary file 2* and *3*.

## Cell lines and culture conditions

The Flp-In T-REx$^{TM}$ HEK293T cell line (purchased from Invitrogen) allows for the generation of stable, doxycycline-inducible expression of transgenes by FLP recombinase mediated integration. This system was used to generate PDE12−/− cell lines, which inducibly expressed PDE12.Strep2.Flag (PDE12.FST2) and the E351A catalytic mutant (E351A.FST2). The E351A point mutation affects a putative magnesium-binding residue and severely compromises the activity of PDE12, as characterised previously (*Rorbach et al., 2011*). In addition, PDE12+/+ and PDE12−/− cell lines were produced which inducibly express mitochondrial poly(A) polymerase (mtPAP). Human embryonic kidney cell lines (HEK293-Flp-In T-REx; HEK293T) were cultured in DMEM (Dulbecco's modified eagle medium) supplemented with 10% (v/v) tetracycline-free fetal bovine serum (FBS), 2 mM Glutamax (Gibco), 1 x Penicillin/Streptomycin (Gibco), 100 µg/ml Zeocin (Invitrogen) and 15 µg/ml Blasticidin (Invivogen) at 37°C under 5% $CO_2$ atmosphere. The cell line was regularly (3–4 months) tested for mycoplasma using LookOut Mycoplasma PCR Detection Kit (Merck). Twenty-four hours prior to transfection cells were split to 10-cm plates in culture medium lacking selective antibiotics, grown to 80–90% confluence and transfected with the transgene-specific pcDNA5/FRT/TO construct and pOG44 using Lipofectamine 2000. Twenty-four hours after transfection, the selective antibiotics hygromycin (100 µg/ml, Invitrogen) and blasticidin (15 µg/ml) were added, and selection medium was replaced every 3–4 days.

To generate the PDE12+/− and PDE12−/− cell lines, HEK293T were transiently transfected to express a pair of CompoZr ZFNs (Sigma-Aldrich), which selectively targeted the *PDE12* gene locus at exon 1 (*Figure 1—figure supplement 1*). HEK293T cells were electroporated with pZFN1 and pZFN2 using Cell Line Nucleofector (Lonza) and buffer kit V (Lonza) applying program A-023. 72 hr following transfection, cells were single-cell cloned into 96-well plates and screened by Sanger sequencing to identify clones harboring indels in *PDE12*. Biallelic modification, and therefore loss of PDE12 protein expression, was confirmed via western blotting.

Human 143B osteosarcoma (HOS) cell line, for the localization of the PDE12.FST2 variant transgenes, was cultured in DMEM containing 2 mM Glutamax, 10% FBS and 1 x Pencillin/Strepomycin (Gibco) at 37°C under 5% $CO_2$ atmosphere. For immunofluorescence experiments, cells were transfected with pcDNA5/FRT/TO-PDE12.FST2 variants using Lipofectamine 2000 according to manufacturer's instructions.

For growth measurements, cells were plated at $1 \times 10^4$ cells/well in 24-well plates and grown with 0.5 ng/ml doxycycline in glucose-free DMEM containing 0.9 g/L galactose, 10% (v/v) FBS, 2 mM

Glutamax and 1 x Penicillin/Streptomycin. Cell confluence was measured at 48 hourly intervals using the IncuCyte HighDefinition Imaging Mode.

## Immunodetection of proteins

The immunofluorescence localization of TOM20 and PDE12.FST2 variants in fixed HOS cells were performed as described previously (*Minczuk et al., 2010*). Immunofluroescence images were captured using a Zeiss LSM 880 confocal microscope.

For immunoblot analysis, 10 µg aliquots of total cell lysate protein or equal volumes of gradient fractions were subjected to SDS-PAGE, wet transferred to PVDF membranes (Millipore), blocked in 5% non-fat milk (Marvel) in PBS for 1 hr and incubated with specific primary antibodies in 5% non-fat milk in PBS overnight. The blots were then incubated with HRP-conjugated secondary antibodies (Promega) in 5% non-fat milk in PBS for 1 hr and visualized using ECL or ECL prime (Amersham).

## [$^{35}$S]-methionine labeling of mitochondrial translation products

To label newly synthesised mtDNA-encoded proteins, growth media was replaced with methionine/cysteine-free DMEM supplemented with 2 mM Glutamax, 110 mg/ml sodium pyruvate and 48 µg/ml cysteine. Cells were incubated for $2 \times 10$ min in this medium before replacement with fresh methionine/cysteine-free DMEM medium containing 10% dialysed FCS and emetine dihydrochloride (Sigma-Aldrich, 100 µg/ml) to inhibit cytosolic translation. Cells were incubated for 20 min before addition of 120 µCi/ml of [$^{35}$S]- methionine (Perkin Elmer). Labelling was performed for 30 min and cells were washed twice with PBS and cells were pelleted. Cells were lysed and aliquots of samples (30 µg) were separated on 10–20% Tris-Glycine SDS-PAGE gels. Gels were dried and products were visualized and quantified with a PhosphorImager system with ImageQuant software.

## Measurement of mitochondrial respiration

Cells were seeded at $4.5 \times 10^4$ cells/well in 200 µl growth medium in XF 24-well cell culture microplates (Seahorse Bioscience) and incubated at 37°C in 5% CO2 for 24 hr. One hour before the assay, growth medium was removed and replaced with assay medium (low buffered DMEM, 10 mM l-glutamine, 1 mM sodium pyruvate, 5 mM galactose), with one wash of assay medium, and left to stabilise in a 37°C non-CO$_2$ incubator. Analysis was performed in quadruplicate using a XF24 Extracellular Flux Analyzer (Seahorse Bioscience). The wells were sequentially injected with 100 nM oligomycin to inhibit ATP synthase, 700 nM carbonylcyanide-4-trifluorometho-xyphenylhydrazone (FCCP) to uncouple the respiratory chain, and 200 nM rotenone to inhibit complex I. Oxygen consumption rate (OCR) was measured for each well every 5 min before and after each injection.

## RNA extraction, mRNA northern blotting

Except where stated, all RNA was extracted from HEK293T cell lines with TRIzol reagent according to manufacturer's instructions.

For northern blots for mt-mRNAs, 5 µg aliquots of total RNA were resolved on 1.2% MOPS-formaldehyde agarose gels (1.2% UltraPure agarose, 0.7 formaldehyde, 1 x MOPS buffer) and transferred to a nylon membrane in 2x SSC buffer. Membranes were hybridized with either radioactively labeled PCR fragments or T7 in vitro transcribed RNA probes corresponding to appropriate regions of mtDNA, exposed to storage Phosphor screens and imaged using a Typhoon scanner.

## Radioactive mitochondrial poly(A) tail (MPAT) assay and MPAT-Seq

This method was adapted from that described by (*Temperley et al., 2003*). 2.5 µg of total RNA was incubated for 15 min at 4°C with 40 pmol of LIGN oligonucleotide, which carries a 3' Spacer C3 modification to prevent concatemerisation. After incubation, ligation was performed in 20 µl reactions using T4 RNA ligase (1x T4 RNA ligase reaction buffer, 10 units T4 RNA ligase (New England Biolabs), 1 mM adenosine triphosphate, two units Turbo DNase, 40 units RNasin RNase inhibitor) at 37°C for 3 hr. Each ligation reaction was subjected to phenol:chloroform extraction followed by sodium acetate/ethanol precipitation of the aqueous phase overnight at −20°C. Ligated RNA was pelleted and resuspended in 10 µl nuclease-free wate. 5 µl aliquots of ligation reaction were subjected to reverse-transcription for 60 min at 37°C using Omniscript reverse-transcriptase (Qiagen) primed with 10 pmol of ANTI-LIGN oligonucleotide. An aliquot of this reaction (12.5%) was used as

template in an initial PCR using 15 pmol each of a gene specific Fw1 and ANTI-LIGN oligonucleotides using KOD polymerase. Primers and free nucleotides were removed after 30 cycles using a QIAquick PCR purification column. Gene-specific Fw2 primers were designed such that when used for nested PCR on the products from the initial PCR, in combination with ANTI-LIGN, a product of exactly 50 bp would be formed for exact cleavage of mt-RNAs with no 3' modification.

For radioactive MPAT assay, Fw2 primers were 5' end labelled with [γ-$^{32}$P]-ATP (Hartmann Analytic) using T4 PNK. A five-cycle nested PCR was performed with 1/6th of the initial PCR purification using 30 nM each of ANTI-LIGN and radiolabelled Fw2 oligos in a 20 µl standard KOD polymerase PCR reaction. Products were resolved on 10% polyacrylamide-urea denaturing gels, gels were dried and exposed to storage Phosphor screen.

For MPAT-Seq, a single PCR was performed with a gene-specific Fw1 and ANTI-LIGN primers for all 22 mt-tRNAs. PCR products were pooled and used as template to produce Illumina sequencing libraries via A tailing and ligation of TruSeq adapters from a TruSeq LT kit.

## Computational analysis of MPAT-Seq data

MPAT primers were removed with cutadapt (*Martin, 2011*) and reads longer than 20nt were mapped using Bowtie2. To ensure that all 3' ends were mapped, reads were aligned to a reference genome based on GRCh38, but with 30 additional, uncalled positions (N) on the 3' end of all mitochondrial rRNAs and tRNAs. To further improve alignment of the 3' ends, mapping parameters '-rdg 7,5 and -rfg 7,5 np 0' were used.

## Circularisation RT-PCR assay

5 µg of total RNA was circularized in a 20 µl reaction using T4 RNA ligase (1x T4 RNA ligase reaction buffer, 10 units T4 RNA ligase (New England Biolabs), 1 mM adenosine triphosphate, two units Turbo DNase, 40 units RNasin RNase inhibitor) for one hour at 37°C. Each ligation reaction was subjected to phenol:chloroform extraction followed by sodium acetate/ethanol precipitation of the aqueous phase overnight at −20°C. Circularized RNA was pelleted at 20,000 *g* for 30 min at 4°C resuspended in 20 µl nuclease-free water. 5 µl aliquots of the circularisation reaction were subjected to reverse-transcription across the circularisation junction for 60 min at 37°C using Omniscript reverse-transcriptase (Qiagen) primed with 10 pmol of gene-specific reverse oligonucleotide, according to manufacturer's instructions. Following reverse transcription, PCR was performed across the ligation junction using gene specific primers Fw1 and Rv1. For the ND1 and COI transcripts, a further nested PCR was performed using the Fw2 and Rv2 oligonucleotides after the initial PCR was purified using QIAquick PCR purification kit. Following PCR, products were cloned using the Zero Blunt TOPO PCR Cloning Kit for Sequencing. Clones were sequenced using M13 Forward universal sequencing primer.

## RNAseq

For analysis of mitochondrial mt-mRNAs via RNAseq, mitochondria were first isolated according to (*Minczuk et al., 2011*). Mitochondrial pellets were resuspended in QIAzol (Qiagen) and RNA was extracted using the miRNeasy kit with on column DNase treatment (Qiagen).

Library generation was performed using the TruSeq Stranded Total RNA LT kit (Illumina, UK) according to manufacturer's instructions. As part of library generation, cytoplasmic and mitochondrial ribosomal RNAs were removed using RiboZero Gold. Quality of libraries was assessed with a D1000 Screentape for TapeStation (Agilent). Libraries were subjected to high-throughput sequencing using the Illumina MiSeq platform.

## Computational RNAseq data analysis

Quality trimming and 3'end adaptor clipping of sequenced reads were performed with Trim_Galore!. Only reads for which pair-ended reads were both longer than 20 nt were mapped to GRCh38 using bowtie2 (*Langmead and Salzberg, 2012*). As cytosolic and mitochondrial ribosomal RNAs were depleted using RiboZero Gold (Illumina), reads that mapped to 12S or 16S mt-rRNAs were removed using Samtools (*Li et al., 2009*). Samtools was also used to create strand-specific files for each sample. Rsubread was used to generate count tables (*Liao et al., 2013*) and DESeq2 was used for statistical analysis of differential gene expression (*Love et al., 2014*).

## Mitochondrial ribosome profiling (MitoRibo-Seq)

For MitoRibo-Seq, parental HEK293 and PDE12−/− cells were treated with 100 µg/ml chloramphenicol and 100 µg/mL cycloheximide for 5 min, rinsed with 5 ml of ice-cold PBS, the dishes were submerged in a reservoir of liquid nitrogen for 10 s and transferred to dry ice. Cells were lysed in 400 µl of lysis buffer (20 mM Tris-HCl (pH 7.5), 150 mM NaCl, 5 mM MgCl$_2$, 1 mM DTT, 1% Triton X-100, 100 µg/ml chloramphenicol (Alfa Aesar) and cycloheximide (Sigma Aldrich)). Cells were scraped and lysates clarified by centrifugation for 20 min at 13,000 $g$ at 4°C.

200 µl aliquots of cell lysates were treated with 7.5 µl RNase 1 (100 units/µl) at 28°C. The reaction was layered onto a 10–30% sucrose gradient as described above for 'Analysis of mitochondrial ribosome integrity'. Subsequently, all fractions containing mitochondrial monosomes, ascertained by western blotting, were pooled and digested with proteinase K (200 µg/ml) for 30 min at 42°C. RPFs were recovered by extracting twice with pre-warmed (65°C) acidic phenol:chloroform and once with chloroform (1:1, v/v, buffered with 10 mM Tris pH 7.5, 0.1 mM EDTA) followed by ethanol precipitation.

RPFs (1 µg) were separated on 15% denaturing polyacrylamide gels and RNA species migrating between 25 and 35 nt were harvested. RNA was eluted from the gel slices and ethanol precipitated. The RNA samples from above were heated at 80°C for 2 min, cooled and the 3' phosphate group removed using T4 polynucleotide kinase (New England Biolabs) for 2 hr at 37°C in a 20 µl reaction. The RNA was concentrated by ethanol precipitation, resuspended in 10 mM Tris-HCl (pH 7.5) and ligated in a 20 µl reaction overnight at 14°C to a preadenylated 3'-adaptor using T4 RNA Ligase 2 (New England Biolabs). RNA was precipitated, loaded into a 15% denaturing polyacrylamide gel and ligated RNA fragments migrating between 48 and 54 nt were excised. The RNA was eluted, precipitated, resuspended as above, and 5' phosphorylated using T4 PNK in the presence of 1 mM ATP for 2 hr at 37°C. RNA was concentrated by ethanol precipitation, resuspended in 10 mM Tris-HCl (pH 7.5) and ligated to a 5' RNA adaptor overnight at 14°C using T4 RNA Ligase (New England Biolabs). The fully adapted RNAs were recovered by ethanol precipitation, dissolved in 6 µl 10 mM Tris-HCl (pH 7.5), and ligated product annealed to an RT primer for 5 min at 65°C. The RNA was subsequently reverse transcribed for 50 min at 55°C in using SuperScript III according to manufacturer's instructions, followed by heat inactivation for 5 min at 85°C. Standard PCR reactions were used to prepare amplicons using forward primer RP1 and RPIX as reverse primer, where X is primer number (1-24). PCR reactions were loaded onto 10% non-denaturing polyacrylamide-TBE gels and products of ~150 bp were excised from the gel and eluted. These amplicon libraries were ethanol precipitated and resuspended in 15 µl 10 mM Tris-HCl (pH 7.5).

Ribosomal RNA was depleted from MitoRibo-Seq samples at the library amplicon stage; 12 µL of the relevant MitoRibo-Seq library was mixed with 4 µL of 4 × hybridization buffer (200 mM HEPES pH 7.5 and 2 M NaCl) and denatured at 98°C for 2 min. DNA was re-annealed for 5 hr at 68°C prior to addition of 2 µL of 10 × DSN master buffer and 2 µL of DSN (four units, Evrogen). Digestion was allowed to proceed for 25 min at 68°C, before addition of 20 µL 10 mM EDTA and incubation for a further 5 min at 68°C. DNA was recovered by a single extraction with phenol–chloroform (1:1, vol/vol) followed by ethanol precipitation and resuspended in 4 µL 10 mM Tris–HCl pH 7.5. The treated amplicon library was subjected to another round of PCR (as above) and the resulting library subjected to a second round of DSN treatment.

Resultant Illumina libraries were sequenced using an Illumina NextSeq 500 platform

## Computational analysis of MitoRibo-Seq data

Adaptor sequences were trimmed using the FASTX-Toolkit, trimmed reads mapping to rRNA were discarded, and remaining reads were mapped to mitochondrial mRNAs using bowtie version 1 (*Langmead et al., 2009*) with parameters -v 2 –best (i.e. maximum two mismatches, report best match). Histograms in *Figure 3C* and *Figure 3—figure supplement 1B* show the positions of the 5' ends of mtRPFs with a + 17 nt offset to map the approximate A-site, and smoothed with a 3-nt running-mean filter. To facilitate precise identification of the A-site, only mtRPFs with lengths in the range 31 to 35 nt and with 5' ends mapping to the first or third positions of codons were used for the codon occupancy plots (*Figure 3B* and *Figure 3—figure supplement 1A*), and the A-site codon was assumed to occupy nucleotide positions 16–18 or 17–19, respectively, of such mtRPFs. These offsets were determined based on analyzing pausing peaks associated with lysine codons in the

ΔFLP mutant, as a function of mtRPF length. Regions of CDS that overlap (ATP8/ATP6 and ND4L/ND4) were excluded due to ambiguous CDS assignment, and mtRPFs with 5' ends mapping within 15 nt of the start codon or 45 nt of the stop codon were excluded. Within each mt-mRNA CDS, mtRPF histograms were scaled by the mean density of mtRPFs within that CDS. Then the scaled density at each A-site codon was summed over all occurrences of that codon species and downweighted by the number of occurrences of that codon species in the mt-mRNA CDS regions examined.

## Analysis of mitochondrial ribosome integrity on density gradients

Cells were lysed in lysis buffer (50 mM Tris (pH 7.4), 150 mM NaCl, 1 mM EDTA, 1% Triton X-100, 20 mM MgOAc). 800 µg of total cell lysates were loaded onto a linear sucrose gradient [2 ml 10–30% (v/v)] in 50 mM Tris-HCl (pH7.2), 20 mM Mg(OAc)$_2$, 80 mM NH$_4$Cl, 0.1M KCl, 1 mM PMSF and centrifuged for 2 hr 15 min at 100,000 $g_{max}$ at 4°C (39,000 rpm, Beckman Coulter TLS-55 rotor). Twenty fractions (100 µl) were collected. 10 µl were used for western blotting and 90 µl was used for extraction of RNA using TRIzol LS reagent according to manufacturer's instructions.

## Aminoacylation assays

Total RNA was extracted using TRIzol reagent according to manufacturer's instructions, with the final pellet resuspended in 10 mM NaOAc at pH 5.0 and kept at 4°C to preserve the aminoacylation state. For deacylated control, the pellet was first resuspended in 200 mM Tris-HCl at pH 9.5 and incubated at 75°C for 5 min, followed by RNA precipitation and resuspension in 10 mM NaOAc at pH 5.0. 5 µg RNA was mixed with an equal volume of sample buffer (0.1M NaOAc, pH 5.0, 8M urea, 0.05% bromophenol blue, 0.05% xylene cyanol) and separated on a 6.5% polyacrylamide gel (19:1 acrylamide:bisacrylamide) containing 8 M urea in 0.1 M NaOAc, pH 5.0. The gel was run at 4°C at 150 V for 1 hr, and then subsequently increased to 500 V until the bromophenol blue reached the bottom of the gel (~16 hr). The portion of the gel between the xylene cyanol and bromophenol blue was dry-blotted onto a nylon membrane (Hybond). Following UV cross-linking (0.120 J), the membrane was hybridized with appropriate radiolabelled riboprobes, and imaged with a PhosphorImager.

Periodate oxidation was performed by incubating total RNA with 50 mM NaIO$_4$ in 100 mM NaOAc/HOAc (pH 4.8). The samples were incubated at room temperature for 30 min and then quenched by adding glucose to 100 mM followed by incubation at room temperature for 5 more minutes. The RNA preparations were subjected to a G25 spin column (GE Healthcare) and then precipitated with ethanol. For the second step, the precipitated NaIO$_4$-treated sample was spun down at 12,000 $g$ for 15 min at 4°C, the supernatant removed and pellet washed in 70% ethanol, spun down again at 7500 $g$ for 5 min at 4°C, the supernatant removed, then the pellet resuspended in 1 M lysine pH 7.4 for 1 hr at 45°C and then passed again through a G25 spin column. To deacylate tRNAs, samples were resuspended in 50 mM Tris-HCl (pH 9) and incubated at 37°C for 30 min. The RNA samples were ethanol-precipitated and subsequently resuspended in water. RNA preparations not subjected to periodate oxidation were prepared in the same manner except that NaIO$_4$ was replaced with NaCl. The loss or retention of the 3' terminal nucleotide was then determined through PAGE and northern blotting.

## Data availability

Sequencing data for MitoRibo-Seq have been deposited in ArrayExpress (http://www.ebi.ac.uk/arrayexpress) under the accession number E-MTAB-5519.

The RNASeq data was uploaded into GEO under the accession number GSE95351 (https://www.ncbi.nlm.nih.gov/geo/query/acc.cgi?acc=GSE95351).

## Acknowledgements

This work has been supported by the core funding provided by the Medical Research Council, UK (MC_U105697135). PR-G is supported by Fundação para a Ciência e a Tecnologia, Portugal (PD/BD/105750/2014). AEF is funded by a grant from the Wellcome Trust, UK (106207). We are grateful to Prof. Robert Lightowlers and Prof. Eric Schon for donating the ΔFLP cells, and to Dr Roman Szczesny and Prof. Andrzej Dziembowski for providing the mtPAP construct.

# Additional information

## Funding

| Funder | Grant reference number | Author |
|--------|----------------------|--------|
| Medical Research Council | MC_U105697135 | Sarah F Pearce<br>Joanna Rorbach<br>Lindsey Van Haute<br>Aaron R D'Souza<br>Pedro Rebelo-Guiomar<br>Christopher A Powell<br>Michal Minczuk |
| Fundação para a Ciência e a Tecnologia | Portugal, PD/BD/105750/2014 | Pedro Rebelo-Guiomar |
| Wellcome | 106207 | Ian Brierley<br>Andrew E Firth |

The funders had no role in study design, data collection and interpretation, or the decision to submit the work for publication.

## Author contributions

SFP, Conceptualization, Data curation, Formal analysis, Validation, Investigation, Visualization, Methodology, Writing—original draft, Project administration, Writing—review and editing; JR, Conceptualization, Data curation, Formal analysis, Investigation, Methodology, Writing—review and editing; LVH, Data curation, Software, Formal analysis, Investigation, Methodology, Writing—review and editing; ARD'S, Data curation, Formal analysis, Investigation, Visualization, Writing—review and editing; PR-G, CAP, IB, Data curation, Formal analysis, Investigation, Writing—review and editing; AEF, Data curation, Software, Formal analysis, Investigation, Visualization, Methodology, Writing—review and editing; MM, Conceptualization, Resources, Data curation, Formal analysis, Supervision, Funding acquisition, Validation, Visualization, Methodology, Writing—original draft, Project administration, Writing—review and editing

## Author ORCIDs

Sarah F Pearce, http://orcid.org/0000-0003-2950-7078
Joanna Rorbach, http://orcid.org/0000-0002-2891-2840
Aaron R D'Souza, http://orcid.org/0000-0001-7169-8440
Christopher A Powell, http://orcid.org/0000-0001-7501-0586
Ian Brierley, http://orcid.org/0000-0003-3965-4370
Andrew E Firth, http://orcid.org/0000-0002-7986-9520
Michal Minczuk, http://orcid.org/0000-0001-8242-1420

# Additional files

## Supplementary files

• Supplementary file 1. Distribution of non-encoded extensions on the mature 3' terminus of mt-rRNA in control cells (+/+), PDE12−/− cells, and PDE12−/− cells expressing either wild-type PDE12 or PDE12 E351A mutant determined by cRT-PCR.

• Supplementary file 2 . Antibodies and vectors.

• Supplementary file 3. Oligonucleotide sequences. Sequences of DNA oligonucleotides for use in radioactive MPAT assay

## Major datasets

The following datasets were generated:

**Database, license,**

| Author(s) | Year | Dataset title | Dataset URL | and accessibility information |
|---|---|---|---|---|
| Van Haute L, Pearce SF, Michal Minczuk | 2017 | Application of next generation sequencing approaches to assess effects of PDE12 knockout on the mitochondrial transcriptome | https://www.ncbi.nlm.nih.gov/geo/query/acc.cgi?acc=GSE95351 | Publicly available at the NCBI Gene Expression Omnibus (accession no: GSE95351) |
| Firth A , Rorbach J, Minczuk M | 2017 | Analysis of mitochondrial gene expression with Ribosome Profiling | http://www.ebi.ac.uk/arrayexpress/experiments/E-MTAB-5519 | Publicly available at ArrayExpress (accession no. E-MTAB-5519) |

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
