## [Decision Letter]

Thank you for submitting your article "Deadenylation specifies a human mitochondrial repair pathway for non-coding RNAs" for consideration by *eLife*. Your article has been reviewed by two peer reviewers, and the evaluation has been overseen by Timothy Nilsen as Reviewing Editor and James Manley as the Senior Editor. The reviewers have opted to remain anonymous.

The reviewers have discussed the reviews with one another and the Reviewing Editor has drafted this decision to help you prepare a revised submission.

While the reviewers and the reviewing editor all felt that the work was interesting and in principle appropriate for *eLife*, several points, some textual and one experimental, need to be addressed via revision.

Summary:

In this manuscript the authors provide evidence to support a model implicating the poly(A)-specific exonuclease PDE12 as a factor responsible for quality control of mitochondrial tRNAs and rRNAs. The authors show that lack of PDE12 results in reduced translation of mitochondrial mRNAs, without affecting the amounts or the polyadenylation of mitochondrial mRNAs or the amounts of 55S monosomes, and without a reduction in the amounts of mitochondrial mRNAs loaded on the ribosomes. Ribosome profiling indicated that translation was stalled in PDE12−/− cell lines at lysine AAG and to a lesser extent at lysine AAA codons. Analysis of the 3' ends of tRNAs by MPAT (mitochondrial poly(A) tail assay) indicated a preponderence of oligoadenylated tRNAs of certain species in PDE12−/− cell lines including tRNA(Lys), tRNA(Ser2) and tRNA(His), but not several others, and analysis of charging by acid Northerns suggested reduced charging levels of tRNA(Lys) and tRNA(Ser2). Control experiments demonstrate that PDE12−/− cell lines have additional oligoadenylation of 16S rRNA, but not 12S rRNA, and separate experiments provide evidence that LSU's and monosomes have an identical amount of oligoadenylation in their 16S rRNA, suggesting that they form normal mitochondrial ribosomes, leading the authors to conclude that the translation defect in PDE12 −/− cells is due to reduced tRNA function. Finally the authors provide evidence that overexpression of mitochondrial PAP causes reduced mitochondrial translation PDE12 +/+ cells, and an extreme reduction of translation in PDE12−/− cells.

The authors conclude that PDE12 has a role in deadenylating promiscuous polyadenylation of mitochondrial tRNAs to improve tRNA function.

Essential revisions:

1) It was agreed that the term "repair" was not an appropriate description for the phenomena you have investigated. A more accurate title would be "Maturation of some human mitochondrial tRNAs requires deadenylation". The term "repair" should be deleted throughout the text.

2) The method that the authors use to measure oligoadenylation (by the MPAT assay) in Figure 4, or by MPAT-seq (Figure 4) requires readthrough of several modified bases that are very difficult to read through. This will result in overestimation of oligoadenylation of unmodified tRNAs relative to that of mature tRNAs, and the resulting assessment of the amount of oligoadenylation will not reflect the total pool of each tRNA species. Moreover, the resulting oligoadenylation measurements will also be differently mis-estimated for each tRNA species, depending on the nature of the corresponding primer extension block or pause, and possibly by different modification levels of each tRNA.

It was agreed that repeating all of the polyA measurements using a different methodology would be onerous. Nevertheless, softening the conclusions reached by using the MPAT assay is required in the context of the potential problems with the assay as noted above.

3) The charging assays in Figure 4 are difficult to interpret because of the assay, which relies on reduced mobility of charged tRNAs relative to uncharged tRNAs, but is confounded in this case by olgoadenylated tRNAs, which also have reduced mobilities. Thus, it is difficult to measure the amount of charging by this assay. The authors should instead measure charging using the method developed by the Pan lab based on periodate treatment.

Related to these issues, it would help if the authors would measure total tRNAs by a Northern as opposed to an acid Northern. If they did this, they could assay total species by hybridization, at single nucleotide resolution.

4) The authors have not provided any information about the mutant PDE12 (E351A). Perhaps they should mention that this mutant was characterized in their previous paper (2011).

5) It was noted that pre-tRNAs with structural impairments may be polyadenylated by the TRAMP complex, followed by 3'-digestion by the nuclear exosome. In this way, polyadenylation functions in tRNA quality control. Could polyadenylation play a related role in mitochondrial, i.e. could PDE12 be prevented from acting on incorrect folded tRNA molecules, which would then be taken out of circulation?

---

## [Author Response]

*Essential revisions:*

*1) It was agreed that the term "repair" was not an appropriate description for the phenomena you have investigated. A more accurate title would be "Maturation of some human mitochondrial tRNAs requires deadenylation". The term "repair" should be deleted throughout the text.*

We have edited the manuscript underscoring the role of PDE12 in the maturation of mitochondrial non-coding RNAs, rather than repair. However, we added a note to the Discussion section on a potential role of PDE12 in mt-tRNA repair after aberrant adenylation (subsection “The role of deadenylation in the biogenesis of mitochondrial tRNA”).

*2) The method that the authors use to measure oligoadenylation (by the MPAT assay) in Figure 4, or by MPAT-seq (Figure 4) requires readthrough of several modified bases that are very difficult to read through. This will result in overestimation of oligoadenylation of unmodified tRNAs relative to that of mature tRNAs, and the resulting assessment of the amount of oligoadenylation will not reflect the total pool of each tRNA species. Moreover, the resulting oligoadenylation measurements will also be differently mis-estimated for each tRNA species, depending on the nature of the corresponding primer extension block or pause, and possibly by different modification levels of each tRNA.*

*It was agreed that repeating all of the polyA measurements using a different methodology would be onerous. Nevertheless, softening the conclusions reached by using the MPAT assay is required in the context of the potential problems with the assay as noted above.*

We agree with the comment that detection of oligoadenylation of post-transcriptionally modified mt-tRNAs by MPAT and MPAT-Seq is likely to be hampered by the reverse transcriptase pausing on the modification sites. We edited the Results and Discussion sections as suggested (please see also our response to point 3, related to measuring total tRNAs by a Northern).

*3) The charging assays in Figure 4 are difficult to interpret because of the assay, which relies on reduced mobility of charged tRNAs relative to uncharged tRNAs, but is confounded in this case by olgoadenylated tRNAs, which also have reduced mobilities. Thus, it is difficult to measure the amount of charging by this assay. The authors should instead measure charging using the method developed by the Pan lab based on periodate treatment.*

*Related to these issues, it would help if the authors would measure total tRNAs by a Northern as opposed to an acid Northern. If they did this, they could assay total species by hybridization, at single nucleotide resolution.*

We thank reviewers for this comment. We agree that the charging assay as per Figure 4 is confounded by the presence of olgoadenylated mt-tRNAs co-migrating with the aminoacylated form of mt-tRNA, nonetheless it clearly shows that the levels of aminoacylated mt-tRNA^Lys^ is reduced. Following the reviewers’ recommendation we have performed additional northern blot analysis on mt-tRNA, including the analysis of mt-tRNA^Lys^ aminoacylation using periodate oxidation (Figure 4—figure supplement 2 and Figure 4—figure supplement 3, respectively). The additional data presented show that oligoadenylation of mt-tRNA leads to reduced levels of mature mt-tRNA^Lys^ available for aminoacylation (Figure 4—figure supplement 2), while the total steady-state levels (mature and oligoadenylated) of this mt-tRNA remains unchanged (Figure 4—figure supplement 2). The periodate-based assay confirmed that only the correctly matured mt-tRNA^Lys^ is aminoacylated (Figure 4—figure supplement 3). We have reflected these observations in the revised manuscript text.

*4) The authors have not provided any information about the mutant PDE12 (E351A). Perhaps they should mention that this mutant was characterized in their previous paper (2011).*

We have added the following information to the Materials and methods section: “The E351A point mutation affects a putative magnesium-binding residue and severely compromises the activity of PDE12, as characterised previously (Rorbach et al., 2011).”

*5) It was noted that pre-tRNAs with structural impairments may be polyadenylated by the TRAMP complex, followed by 3'-digestion by the nuclear exosome. In this way, polyadenylation functions in tRNA quality control. Could polyadenylation play a related role in mitochondrial, i.e. could PDE12 be prevented from acting on incorrect folded tRNA molecules, which would then be taken out of circulation?*

We have added a comment explaining the role of the TRAMP polyadenylation complex in the digestion of aberrantly processed non-coding RNAs, including tRNAs in the eukaryotic nucleus. As we do not observe a reduction in overall steady-state level of mt-tRNAs in the absence of PDE12, determined by combined quantification of mature and oligoadenylated species of mt-tRNAs pools (Figure 4—figure supplement 4C), we do not believe that the oligoadenylation observed functions as a signal for the turnover of mt-tRNA species.